# The actions of others act as a pseudo-reward to drive imitation in the context of social reinforcement learning

Anis Najar[1,2,3]*, Emmanuelle Bonnet[1,2,3], Bahador Bahrami[4,5,6], Stefano Palminteri[1,2,3]*

**1** Laboratoire de Neurosciences Cognitives et Computationnelles, Institut National de la Santé et de la Recherche Médicale, Paris, France, **2** Département d'Études Cognitives, École Normale Supérieure, Paris, France, **3** Human Reinforcement Learning team, Université de Paris Sciences et Lettres, Paris, France, **4** Ludwig-Maximilians Universität München, Faculty of Psychology and Educational Sciences, General and Experimental Psychology, Munich, Germany, **5** Department of Psychology, Royal Holloway University of London, London United Kingdom, **6** Max Planck Institute for Human Development, Center for Adaptive Rationality, Berlin, Germany

* anis.najar@ens.fr (AN); stefano.palminteri@ens.fr (SP)

**Data Availability Statement:** https://github.com/hrl-team/mfree_imitation.

**Funding:** SP was supported by an ATIP-Avenir grant (R16069JS), the Programme Emergence(s)

## Abstract

While there is no doubt that social signals affect human reinforcement learning, there is still no consensus about how this process is computationally implemented. To address this issue, we compared three psychologically plausible hypotheses about the algorithmic implementation of imitation in reinforcement learning. The first hypothesis, decision biasing (DB), postulates that imitation consists in transiently biasing the learner's action selection without affecting their value function. According to the second hypothesis, model-based imitation (MB), the learner infers the demonstrator's value function through inverse reinforcement learning and uses it to bias action selection. Finally, according to the third hypothesis, value shaping (VS), the demonstrator's actions directly affect the learner's value function. We tested these three hypotheses in 2 experiments ($N = 24$ and $N = 44$) featuring a new variant of a social reinforcement learning task. We show through model comparison and model simulation that VS provides the best explanation of learner's behavior. Results replicated in a third independent experiment featuring a larger cohort and a different design ($N = 302$). In our experiments, we also manipulated the quality of the demonstrators' choices and found that learners were able to adapt their imitation rate, so that only skilled demonstrators were imitated. We proposed and tested an efficient meta-learning process to account for this effect, where imitation is regulated by the agreement between the learner and the demonstrator. In sum, our findings provide new insights and perspectives on the computational mechanisms underlying adaptive imitation in human reinforcement learning.

## Introduction

The complexity of our society could not have been achieved if humans had to rely only on individual learning to identify solutions for everyday decision problems. An isolated learner must invest sufficient energy and time to explore the available options and may thus encounter

de la Ville de Paris, the Fondation Fyssen and the Fondation Schlumberger pour l'Education et la Recherche (FSER) and the FrontCog grant (ANR-17-EURE-0017) BB was supported by the European Research Council (ERC) under the European Union's Horizon 2020 research and innovation programme (819040 - acronym: rid-O). BB was supported by the NOMIS foundation and the Humboldt Foundation. AN's salary was funded by the ATIP-Avenir. The funders had no role in study design, data collection and analysis, decision to publish, or preparation of the manuscript.

**Competing interests:** The authors have declared that no competing interests exist

**Abbreviations:** AIC, Akaike Information Criterion; DB, decision biasing; IQ, intelligence quotient; LPP, Log Posterior Probability; MB, model-based imitation; RW, Rescorla–Wagner; SD, Skilled Demonstrator; UD, Unskilled Demonstrator; VS, value shaping.

unexpected negative outcomes, making individual learning slow, costly, and risky. Social learning may largely mitigate the costs and risks of individual learning by capitalizing on the knowledge and experience of other participants.

Imitation is one of the main mechanisms of social learning and has been widely investigated in psychology and neuroscience [1,2]. It has been studied from various frameworks, such as the mirror neurons system [3], theory of mind [4], and Bayesian inference [5]. While these studies provide valuable insights about the computational mechanisms of imitation, they are still limited in their scope as imitation is treated in isolation from other learning processes, such as autonomous reinforcement learning. Here, we compare three radically different and psychologically plausible computational implementations of how imitation can be integrated into a standard reinforcement learning algorithm [6–9]. To illustrate the three hypotheses, we consider the stylized situation, where a reinforcement Learner is exposed to the choices of a Demonstrator, before making her own choices.

The first hypothesis, decision biasing (DB), well represented in the cognitive neuroscience literature, postulates that observing a Demonstrator's choice influences the learning process by biasing the Learner's decision on the next trial toward the most recently observed demonstration [6,7]. This implementation presents one limitation in that it does not allow an extended effect of imitation over time. This hypothesis conceptualizes imitation as biasing the exploration strategy of the Learner and contrasts with several observations in experimental psychology suggesting that social signals have long-lasting effects on the Learner's behavior [10]. In addition, the experimental designs of these previous studies were not well suited for assessing whether imitation accumulates over successive demonstrations and propagates over several trials, since one observational trial was strictly followed by one private trial.

A second account consists in framing imitation as an inverse reinforcement learning problem, where the Learner infers the preferences of the Demonstrator [5,9]. In previous paradigms, the model of the Demonstrator is generally used to predict her behavior, but these representations could easily be recycled to influence the behavior of the Learner. Unlike the DB account, this model-based imitation (MB) allows for the accumulation of demonstrations and the propagation of imitation over several trials. However, this method is computationally more demanding and still leaves unanswered the question of how the model of the Demonstrator is integrated into the behavior of the Learner.

In line with previous works on advice taking [11] and learning from evaluative feedback [12], we propose a third approach where the demonstrations influence the value function instead of option selection [13]. As such, this value shaping (VS) scheme has the desirable property of allowing for a long-lasting influence of social signals while being computationally simple.

We compared these three computational implementations of imitation (and a baseline model without social learning) against empirical data from two independent experiments (Exp 1: $N = 24$, Exp 2: $N = 44$). Both experiments featured a new variant of a social reinforcement learning task, where, to assess the accumulation and the propagation of social signals, we manipulated the number of demonstrations and private choices in a trial-by-trial basis (Fig 1). We also tested the robustness and generalization of the results in a third, independent, experiment featuring a larger sample size ($N = 302$) and different design specifications [14]. We also manipulated the skills of the Demonstrator to assess whether the imitation parameters were modulated in an adaptive manner. This manipulation was implemented within-subject in Exp 1 and between-subject in Exp 2.

Overall, quantitative model comparison indicated that imitation, whenever adaptive (i.e., when the Demonstrator outperforms the Learner), takes the computational form of VS, rather than DB or MB.

We also analyzed the parameters of the winning model to determine whether or not, in the context of social reinforcement learning, more weight is given to information derived from

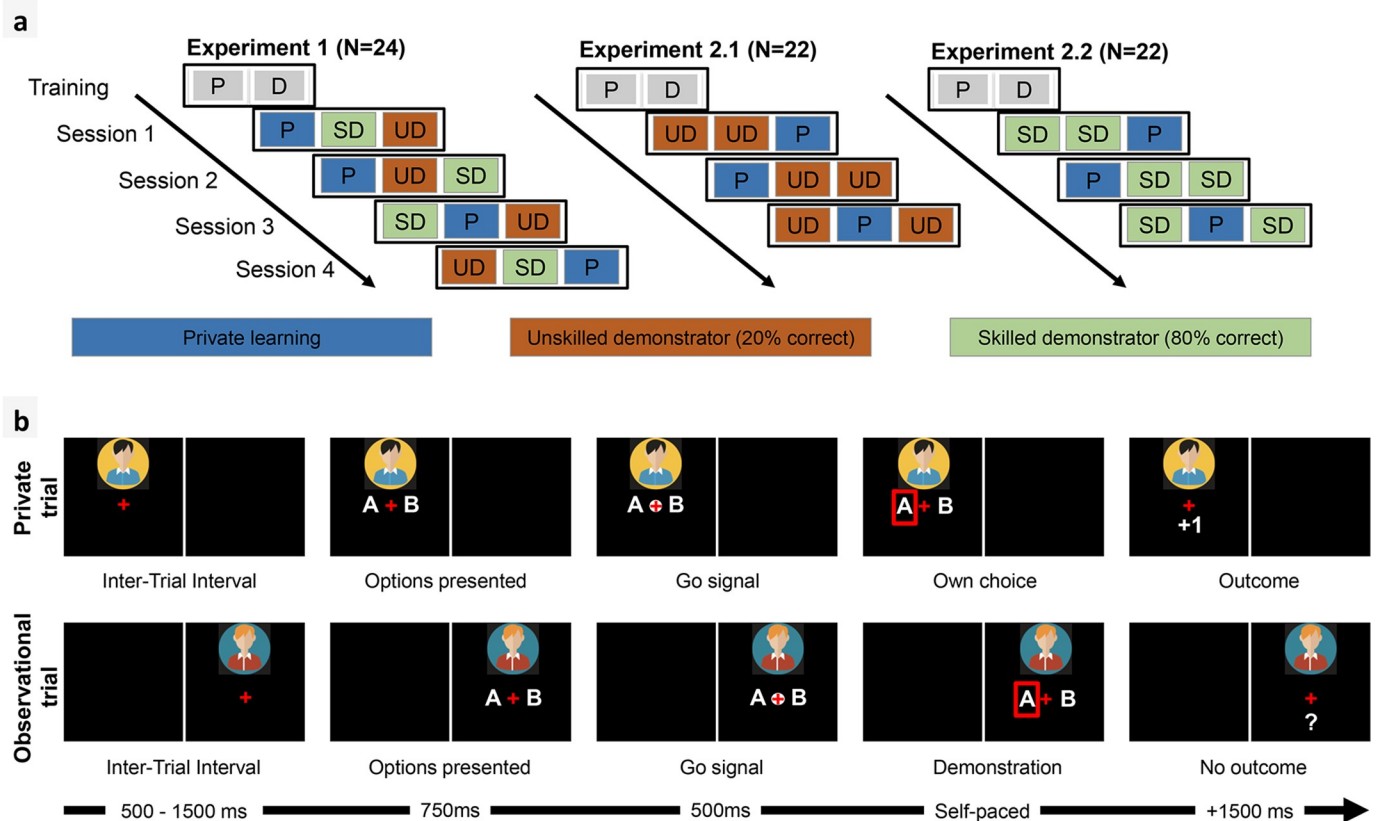

**Fig 1. Experimental design.** (a) Experimental conditions. Conditions were randomized within sessions. In the Private condition (P), there was no social information, and the task was a basic reinforcement learning task. In observational conditions (D), the Learner observed choices of another player that were interleaved with her own choices. The training involved a short Private session followed by an Observational session. In the SD condition, the other player picked the reward maximizing symbol 80% of the time, while in the UD condition, the other player picked the reward maximizing symbol only 20% of the time. The correct choice rate of the other player (SD–UD contrast) was manipulated within-subject in Exp 1 and between-subjects in Exp 2. Experiments 1 and 2 comprised 4 and 3 sessions, respectively. (b) Typical observational and private trials (timing of the screen given in milliseconds). During observational trials, the Learner was asked to match the choice of the other player before moving to the next trial. The Demonstrator's outcome was not shown and replaced by a question mark "?". P, private; SD, Skilled Demonstrator; UD, Unskilled Demonstrator.

oneself compared to the other [15]. The comparison of the private reward learning rate with the imitation learning rate was overall consistent with more weight given to privately generated information. Finally, we compared the imitation learning rates across different observational conditions (Skilled Demonstrator (SD) versus Unskilled Demonstrator (UD)) to see whether imitation is modulated. Consistent with previous theoretical and empirical work about meta-learning [8,16], we found that participants can infer the skill of a Demonstrator and regulate imitation accordingly. This result was even more pronounced when the skill of the Demonstrator was implemented within-subject, which suggests that being exposed to both SD and UD models could be more effective in preventing Learners from bad influence. We formalized this intuition in a meta-learning model where imitation is flexibly modulated by the inferred Demonstrator's skills.

## Results

### Experimental design

We performed 2 experiments implementing a probabilistic instrumental learning task (Fig 1), where the participants were repeatedly presented with a binary choice between 2 abstract

visual stimuli resulting in either winning or losing a point. Symbols presented together in a choice context had opposite and reciprocal winning and losing probabilities (0.7/0.3 in Exp 1 and 0.6/0.4 in Exp 2). The goal of the participant, i.e., the Learner, was to learn by trial-and-error which of the 2 stimuli had the highest expected value. Three learning conditions were presented in blocs of 20 trials each per session. In the Private condition (P), no information other than reward outcome was provided to the Learner. Observational conditions involved additional social information that was randomly interleaved within private choices. Specifically, during observational trials, the Learner could observe the choice of another player, hereafter referred to as the Demonstrator, on the same pair of stimuli. The outcome of the Demonstrator was never shown as we were interested in pure imitation and not vicarious reinforcement learning. Participants were recruited in pairs and were told that they were observing the choices of the other participant. However, the demonstrations were controlled by the computer and drawn from 2 Bernoulli distributions. In the SD condition, the other player picked the reward maximizing stimulus 80% of time. In the UD condition, the other player picked the reward maximizing stimulus only 20% of time. The SD/UD manipulation was within-subject in Exp 1 (all participants were exposed to both SD and UD) and between-subject in Exp 2 (each participant was exposed to either the SD or the UD).

## Behavioral analyses

As a quality check, we first assessed whether or not participants were able to identify the correct (i.e., reward maximizing) options. We found the correct choice rate to be significantly above chance level in both experiments (Exp 1: M = 0.75, SD = 0.08, t(23) = 14.85, p = 2.8e-13; Exp 2: M = 0.62, SD = 0.09, t(43) = 8.56, p = 7.69e-11), indicating that, in average, participants identified the reward maximizing options. We then assessed whether or not the correct choice rate was affected by the skill of the Demonstrator (S1 Fig). In Exp 1, even if as expected the average correct choice rate was higher in the SD compared to the UD condition (76% versus 73%), the difference did not reach statistical significance ($F(2,69) = 0.426$, $p = 0.655$). In Exp 2, we found a significant interaction between Demonstrator's performance and social information ($F(1,84) = 4.544$, $p = 0.036$). The effect was driven by Learners in the SD group having a correct choice rate differential greater than 0 (+5%) comparing the Observational and Private conditions and Learners in the UD having a differential smaller than 0 (−4%) (direct comparison of the learning differential between UD versus SD: Welch 2-sample $t$ test $t(35.215) = 2.8589$, $p = 0.007$). These results indicate that Demonstrators' skill affected Learners' performance.

## Model comparison

We fitted 4 computational models to the behavioral data of both experiments (Fig 2A). In our model space, the baseline was represented by a Rescorla–Wagner (RW) model that does not integrate social information into the learning process (i.e., it treats Private and Observational conditions equivalently). The second model, DB, assumes that demonstrations only bias action selection on the next trial [6]. The third model, MB, implements an inverse reinforcement learning process that infers a model of the Demonstrator's preferences and uses it for biasing action selection.

Finally, in the VS model, Demonstrator's choices directly affect the value function of the Learner using the following equation:

$$Q(d) \leftarrow Q(d) + \alpha_i \times [1 - Q(d)], \tag{1}$$

where $d$ is the action chosen by the Demonstrator, $Q$ is the value function of the *Learner*, and $\alpha_i$ is an imitation learning rate. In other terms, the VS model assumes that the Demonstrator's choice is perceived as a positive outcome (or surrogate reward).

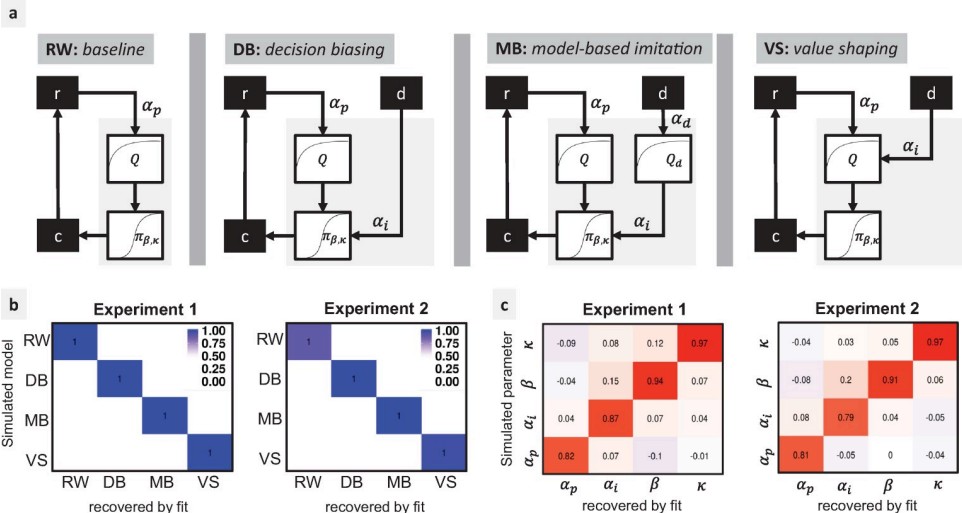

**Fig 2. Model comparison.** (a) Model space: RW: The baseline is a Rescorla-Wagner model that only learns from self-experienced rewards. Rewards $r$ are used for computing a value function $Q$, which, in turn, is used for deriving a policy $\pi$ using a softmax function. Participant choices $c$ are sampled according to $\pi$. The demonstrations $d$ have no effect on the learning process. DB: Demonstrations are used for biasing the participant's policy without affecting the value function. MB: Demonstrations are used for inferring the Demonstrator's value function $Q_d$, which is then used for biasing the participant's decisions. VS: Demonstrations directly affect the participant's value function. $\alpha_p$ private reward learning rate. $\alpha_i$ imitation decision bias parameter (DB, MB models) or imitation learning rate (VS model). $\alpha_d$ demonstration learning rate. $\beta$ softmax inverse temperature. $\kappa$ choice autocorrelation. (b) Model recovery: model frequencies (blue shade) and exceedance probabilities (XP = 1) for each pair of simulated/fitted models, based on the AIC. (c) Parameter recovery: Spearman correlation scores between each pair of simulated/fitted parameter values for the winning model VS. AIC, Akaike Information Criterion; DB, decision biasing; MB, model-based imitation; RW, Rescorla–Wagner; VS, value shaping.

The exact computational implementation of each model was first optimized independently, by comparing several alternative implementations of the same model: 2 implementations of RW, 6 for DB, 9 for MB, and 2 for VS (Methods). In the main text, we only report the comparison between the best implementations of each model. The intermediate model comparison results are provided with the Supporting information (cf. S2 Fig).

We first performed a model recovery analysis to check if the models are identifiable within our task and parameter space (Fig 2B). We then compared the different models in the empirical data (Fig 3A). In Exp 1, roughly the same proportion of participants were explained by RW (Model Frequency MF: 0.45, Exceedance Probability XP: 0.72) and VS (MF: 0.33, XP: 0.24). These results can easily be explained by the fact that each participant encountered both SD and UD. Since UD should not be imitated, their presence can justify the observed high frequency of RW. We verified this intuition by fitting separately UD and SD conditions (plus the Private condition). Indeed, we observed that participants' behavior in the UD condition was best explained by RW (MF: 1, XP: 1), whereas the choice data in the SD condition were better explained by VS (MF: 0.9, XP: 1).

The same pattern was found in Exp 2 where the skill of the Demonstrator was manipulated between-subjects. The subgroup of participants in the UD condition were best explained by the RW model (MF: 0.63, XP: 0.89), while the subgroup of participants in the SD condition were best explained by the VS model (MF: 1, XP: 1).

We also note that in our model space, parameters penalization only concerns the comparison between the RW versus the all imitation models. As the 3 imitation models have the same number of free parameters, the DB, MB, and VS models can be compared just using the

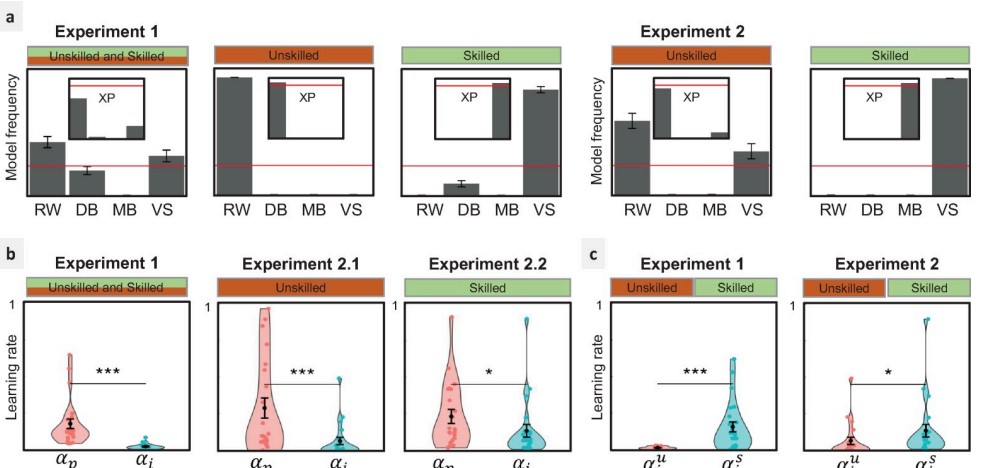

**Fig 3. Results.** (a) Model comparison: model frequencies and exceedance probabilities (XP) of the fitted models based on the AIC. Red lines: 0.25 chance level for model frequency and 0.95 threshold for exceedance probability. (b, c) Learning rates of the winning model VS. (b) Comparison between private ($\boldsymbol{\alpha_p}$) and imitation ($\boldsymbol{\alpha_i}$) learning rates. (c) Comparison between imitation learning rates when observing the UD ($\boldsymbol{\alpha_i^u}$) and the SD ($\boldsymbol{\alpha_i^s}$). $^*p<0.05$, $^{***}p<0.001$, Wilcoxon test. Underlying data can be found in https://github.com/hrl-team/mfree_imitation/. AIC, Akaike Information Criterion; DB, decision biasing; MB, model-based imitation; RW, Rescorla–Wagner; SD, Skilled Demonstrator; UD, Unskilled Demonstrator; VS, value shaping.

maximum likelihood without incurring overfitting. Overall, model comparison results (based on the maximum likelihood comparison restricted to the DB, MB, and VS models) confirm our conclusions that the VS outperforms DB and MB (cf. S10 Fig).

Of note, the model comparison results were robust across different implementations of the model space, notably with or without asymmetric value update in private learning, with or without including a choice autocorrelation parameter, and with or without allowing for negative imitation learning rates (Methods; S3 Fig). To sum up, in both experiments, we consistently found that, when useful, imitation is computationally implemented in a VS way. These results show that in our task, imitation has a direct effect on the value function of the Learner, by reinforcing the observed actions, but without building an explicit model of the Demonstrator.

## Model properties

We analyzed model simulations to identify specific behavioral signatures of the different imitation models. This analysis capitalizes on the specific feature of our experimental design, namely the fact that we allowed for several observational or private trials to be presented in a row (while keeping their overall number the same). This feature allowed us to assess the accumulation and the propagation of social information over several consecutive trials. We restricted the analysis to the SD condition, as we already showed that imitation is suppressed in the UD conditions. We defined the behavioral imitation rates as the average number of trials where the choice $c_i$ was equal to the demonstration $d_j$. Behavioral imitation rates were higher than chance in all cases (cf. S6 Fig). We defined accumulation as the difference in behavioral imitation rates between 2 successive demonstrations $d_{t-1}$ and $d_{t-2}$ preceding a private choice $c_t$. A Learner paying attention only to the last demonstration should display a positive differential. Similarly, propagation was measured as the difference in behavioral imitation rates between 2 successive private trials $c_{t+1}$ and $c_{t+2}$ following a demonstration $d_t$. A Learner that uses demonstrations only to bias exploration should display a positive differential.

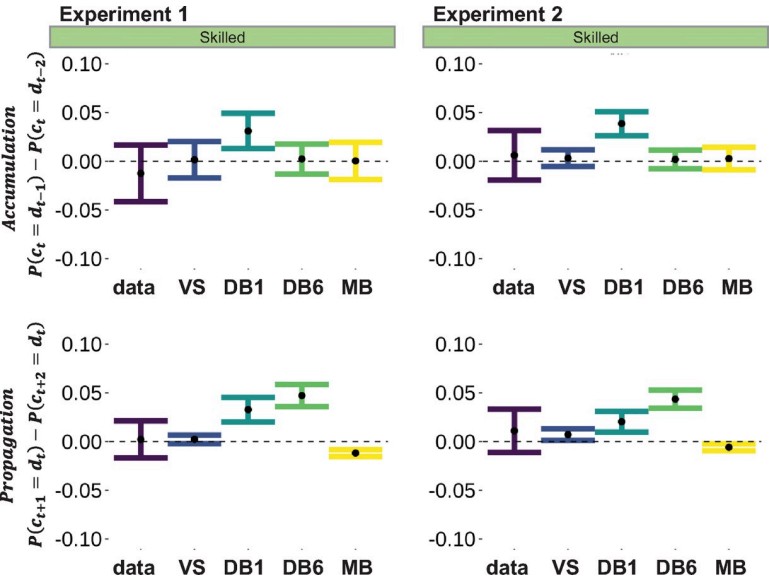

**Fig 4. Model properties.** Difference in the behavioral imitation rate calculated between the 2 preceding demonstrations ($d_{t-1}$ minus $d_{t-2}$; top row) and calculated between 2 consecutive choices ($c_{t+1}$ minus $c_{t+2}$; bottom row) for Exp 1 (left column) and Exp 2 (right column). The first metric measures the accumulation of social signals across consecutive observational trials, and the second metric measures the propagation of social signals across consecutive private trials. Underlying data can be found in https://github.com/hrl-team/mfree_imitation/. DB, decision biasing; MB, model-based imitation; VS, value shaping.

We analyzed these metrics in both empirical and simulated data (Fig 4). Models were simulated using the set of the best fitting parameters. In addition to the VS and MB models, we included 2 variants of the DB model (see Methods). The first variant, DB1, corresponds to the standard implementation of DB that has been widely considered in the literature, which implements a "pure" DB process [6,17]. The second variant, DB6, is our improved version of DB1, which includes the accumulation of demonstrations over consecutive observational trials in a temporary memory trace and was the winning implementation within the DB family of models (cf. S2 Fig).

As expected, we found that the standard model (DB1) failed to capture the effect of accumulation of demonstrations, since the demonstration at $d_{t-2}$ was associated with a smaller behavioral imitation rate compared to the demonstration at $d_{t-1}$. This limitation was corrected by allowing a short-term accumulation of the demonstration in DB6, whose simulations were closer to the real data, to the same extent as VS and MB. However, both DB1 and DB6 were incapable of reproducing the propagation of the demonstration over trials: The private choice at $c_{t+2}$ was associated with a smaller behavioral imitation rate compared to the private choice at $c_{t+1}$. Both VS and MB made good predictions about the propagation effect, but VS was numerically closer to the real data compared to MB.

In summary, we did find different behavioral signatures of the different models in relation to a specific feature of our design: allowing for more than 1 consecutive private or observational trials. All models (except the standard DB: DB1) were capable of reproducing the effect of accumulation. However, neither version of DB was capable of reproducing the effect of propagation. Thus, model simulations "falsify" the DB model and consolidate the model comparison results to support VS as the "winning" model.

Finally, in addition to analyzing these distinctive features of our model given our design, we also checked whether the winning model was capable of capturing the observed behavior in an

unbiased manner, by computing the correlation between observed and simulated data points. We found that the VS model captured well both between-trial and between-subject variance, as all the correlations displayed slopes very close to 1 and intercepts very close to 0 (cf. S7 and S8 Figs).

## Parameter analysis

After verifying that the parameters of the winning model, VS, can be properly identified (Fig 2C), we analyzed the fitted parameters across experiments and conditions (Fig 3B and 3C).

We first assessed whether participants put more weight on their freely obtained outcomes or on the choices of the Demonstrator. We found the private reward learning rates were significantly larger compared to the imitation learning rates (Fig 3B). This was true in Exp 1 (Wilcoxon signed-rank test: $V = 300$, $p = 1.192e{-}07$). In Exp 2, the difference between private and imitation learning rates was more pronounced when confronted with a UD (Wilcoxon signed-rank test: $V = 251$, $p = 1.431e{-}06$) and still detectable when facing an SD (Wilcoxon signed-rank test: $V = 200$, $p = 0.01558$).

We then compared imitation learning rates across different observational conditions to see whether imitation was modulated by the skill of the Demonstrator (Fig 3C). We found the imitation learning rate significantly higher in the SD condition compared to the UD condition (Exp 1: Wilcoxon signed-rank test: $V = 9$, $p = 3.934e{-}06$; Exp 2: Wilcoxon rank-sum test: $W = 137$, $p = 0.01312$). We also inspected the other (nonsocial) parameters such as choice inverse temperature and private reward learning rate to see whether the effect of the Demonstrator skill was specific to imitation. This analysis revealed no significant difference between the nonsocial parameters fitted in the UD and the SD conditions (Exp 1: all $p{>}0.09$; Exp 2: all $p{>}0.2$; cf. S4 Fig). This suggests that the strategic adjustment induced by the Demonstrator's skill is specific to the imitation process. The result of this analysis is consistent with the model comparison results indicating that different computational models (RW and VS) explain the Learner's behavior in the different observational conditions (UD and SD, respectively), as a result of a meta-learning process where the skill of the Demonstrator is inferred to modulate imitation.

## Generalizing the results

To test the robustness of our results in another experimental setting and a larger cohort, we analyzed an additional dataset ($N = 302$) from a recently published study [14]. This experiment, hereafter noted as Exp 3, differs from ours in several aspects (see Methods). First of all, the outcome contingencies were not static, but followed a random walk (Fig 5A), and Demonstrator choices were actual choices recorded from 2 previous participants playing the exact same reward contingencies. One of the participants used as Demonstrator displayed relatively poor skill ($0.60 \pm 0.05$), while the other relatively good skill ($0.84 \pm 0.03$). The same Demonstrator choices were not necessarily displayed across participants. As a result, the observed Demonstrator's correct choice rate ranged from 0.53 to 0.92. First, we checked that using Exp 3 set up we got a good model recovery with respect to our main model space, allowing us to meaningfully compare the RW, MB, DB, and VS models (Fig 5B). Second, we compared the models on the empirical data and, replicating Exp 1 and Exp 2 results, we found that the VS model was the winning one (MF: 0.53, XP: 0.97, Fig 5C).

We then assessed whether participants weighted their freely obtained outcomes more than the choices of the Demonstrator as they did in Exp 1 and Exp 2. Again, we found the private reward learning rates significantly higher compared to the imitation learning rates (Wilcoxon signed-rank test: $V = 42747$, $p{<}2.2e{-}16$). Finally, Exp 3 presented a novel feature such that half of the participants were informed about the intelligence quotient (IQ; approximated by

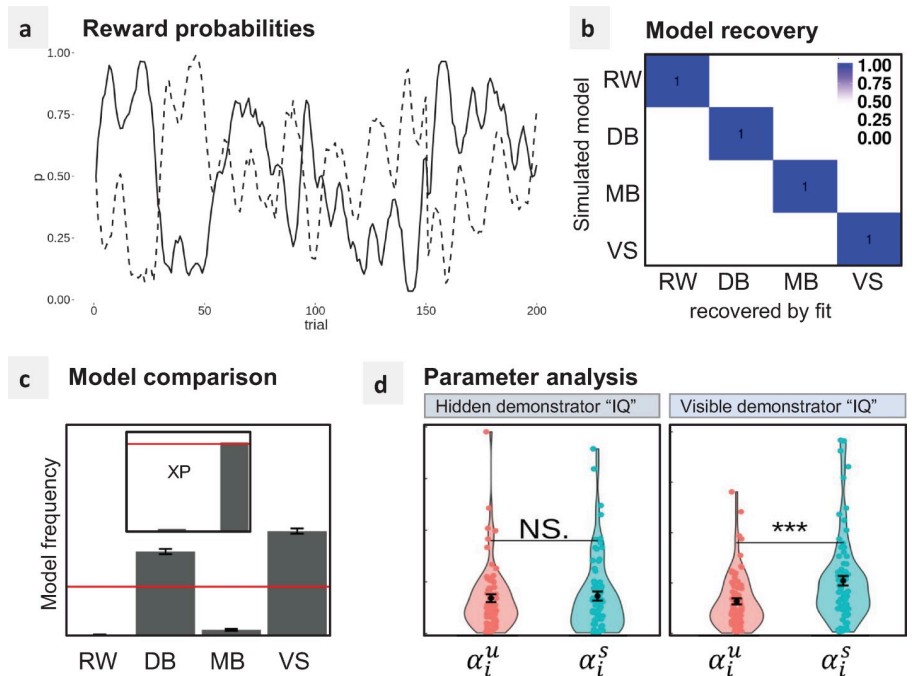

**Fig 5. Generalizing the results.** (a) Reward probabilities: In Exp 3, outcome contingencies for both stimuli were not static, but followed a random walk (restless 2-armed bandit task). All participants were exposed to the same contingencies. (b) Model recovery: model frequencies (blue shade) and exceedance probabilities (XP = 1) for each pair of simulated/fitted models, based on the AIC. (c) Model comparison: model frequencies and exceedance probabilities (XP) of the fitted models based on the AIC. Red lines: 0.25 chance level for model frequency and 0.95 threshold for exceedance probability. (d) Parameter analysis: learning rates of the winning model VS. Left: comparison between private ($\alpha_p$) and imitation ($\alpha_i$) learning rates. Right: comparison between imitation learning rates when observing the UD ($\alpha_i^u$) and the SD ($\alpha_i^s$). ***$p<0.001$, Wilcoxon test. Underlying data can be found in https://github.com/hrl-team/mfree_imitation/. AIC, Akaike Information Criterion; DB, decision biasing; IQ, intelligence quotient; MB, model-based imitation; NS, not significant; RW, Rescorla–Wagner; SD, Skilled Demonstrator; UD, Unskilled Demonstrator; RW, Rescorla–Wagner; VS, value shaping.

Raven's progressive matrices score) of their Demonstrator ($N = 150$), while the other half were not ($N = 152$). In this experiment, the SD (0.84±0.03 correct response rate) presented a higher nonverbal IQ (28/30) compared to the UD (0.64±0.05 correct response rate; IQ = 15/30). This configuration allowed us to test whether a smaller gap in the Demonstrator's skills (roughly 20% difference in the correct response rate) is sufficient to induce a detectable modulation in the imitation learning rate and whether the modulation is sensitive to exogenous cues signaling cognitive ability.

We submitted the imitation rate to a 2-way ANOVA with IQ information (visible/hidden) and Demonstrator's skill as a between-subject factors, and we found a main effect ($F(1,298) = 8.501$, $p = 0.00382$) moderated by the IQ visibility factor (interaction: $F(1,298) = 5.248$, $p = 0.02267$). Post hoc test confirmed that the imitation learning rates were significantly different only in the "visible IQ" participants (Wilcoxon rank-sum test: $W = 2092$, $p = 0.0002356$), but not in the "hidden IQ" participants (Wilcoxon rank-sum test: $W = 2634$, $p = 0.7544$) (Fig 5D).

## Modulation of imitation

In order to account for the modulation of imitation as a function of the Demonstrator's skill, we implemented a meta-learning model where Learners assess the performance of the

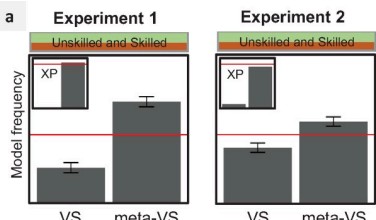 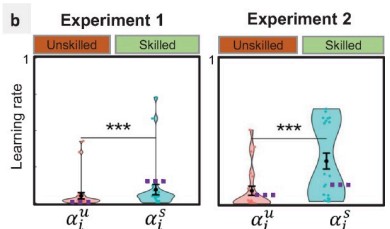

**Fig 6. Meta-learning model. (a)** Model comparison: model frequencies and exceedance probabilities (XP) of the fitted models based on the AIC. Red lines: 0.5 chance level for model frequency and 0.95 threshold for exceedance probability. meta-VS: meta-learning model where imitation is implemented via VS, and the imitation learning rate is dynamically updated. **(b)** Average imitation learning rates of the meta-VS model: comparison between imitation learning rates when observing the UD ($\alpha_i^u$) and the SD ($\alpha_i^s$). The purple dots represent the average imitation learning rate of the VS model. ***$p<$**0.001**, Wilcoxon test. Underlying data can be found in https://github.com/hrl-team/mfree_imitation/. AIC, Akaike Information Criterion; SD, Skilled Demonstrator; UD, Unskilled Demonstrator; VS, value shaping.

Demonstrator and adapt their imitation accordingly. Since the Demonstrator's choice outcomes are not directly accessible to the Learner, we postulated that the participant uses their own value function to assess the Demonstrator's skill on the task. The imitation rate is first initialized to 0 for every state (i.e., pair of symbols), then updated trial by trial using an auxiliary learning rate (see "Methods"). In this framework, if the Demonstrator chooses the option that the Learner currently believes is the best, in other words, if the Demonstrator agrees with the Learner, the imitation rate increases. The converse is true when the Demonstrator chooses the option that the Learner currently believes is not the best. By comparing the agreement between Demonstrator's choices to their own value function, the Learner is able to track a subjective inferred skill of the Demonstrator by relying only on his own evaluation of the task, without building an explicit model of the Demonstrator. This would provide an effective, and yet model-free, way to infer the Demonstrator's skill and regulate imitation accordingly.

We fitted this meta-learning model (which we label meta-VS) and compared it to the VS model. Results show that the meta-VS model explains the participant's choices better than the baseline VS model in Exp 1 (MF: 0.74, XP: 0.99), and Exp 2 (MF: 0.59, XP: 0.9) (Fig 6A). We also analyzed the resulting trial-by-trial estimates of the imitation learning rates averaged separately for SD and the UD conditions and found that we were able to reproduce the observed significant modulation of the imitation rates (Exp 1: Wilcoxon signed-rank test: $V = 0$, $p = 1.192e{-}07$; Exp 2: Wilcoxon rank-sum test: $W = 88$, $p = 0.0001742$) (Fig 6B).

## Discussion

Over 3 experiments, featuring different outcome contingencies and different ratios between private and observational learning trials, we found that imitation takes the computational form of VS, which implies that the choices of the Demonstrator affect the value function of the Learner. On top of that, we found that imitation is modulated by a meta-learning process, such that it occurs when it is adaptive (i.e., SD).

In other terms, imitation is instantiated as a model-free learning process, as it does not require an explicit model of the Demonstrator [18]. Our conclusions are based on model comparison and parameter comparison analyses, whose validity is supported by having verified in simulated data good model and parameter recoveries. Our results are robust across experiments and across different implementations of the computational models. A comparison between reward and imitation learning rates suggests that privately generated outcomes are

overweighted compared to social information. Finally, in accordance with a vast body of evolutionary literature indicating that, in order to be adaptive, imitation should be modulated [19], we found that humans can correctly infer the skill of a Demonstrator and modulate their imitation accordingly [7]. We proposed and tested a simple way to achieve the modulation of imitation, without incurring the computational cost of building an explicit model of the Demonstrator.

## Comparing value shaping to decision biasing

DB postulates that imitation is essentially an exploration bias rather than a learning process. By contrast, VS allows the Demonstrator's choices to have a stronger and long-lasting influence on the Learner's behavior, as demonstrations are progressively accumulated and stored in the value function. A notable advantage of VS is that it can easily account for observational learning in Pavlovian settings where no decisions are involved, while DB cannot [20,21]. Our VS method is equivalent to the outcome bonus method that has been proposed for integrating advice into reinforcement learning [11]. This method stipulates that the advised options are perceived more positively and thus acquire an extra reward bonus. In the reinforcement learning literature, this strategy is called reward shaping, which corresponds to augmenting the reward function with extra rewards in order to speed up the learning process [13,22]. However, it has been widely reported that reward shaping can lead to suboptimal solutions that fail to account for human behavior [12,23,24]. Several solutions have been proposed to address this problem, such as VS which affects the preference for "advised" actions without modifying the Learner's reward specifications [12], and policy shaping which affects the Learner's behavior without modifying its value function (note that policy shaping is different from DB as it has a long-lasting effect) [25,26].

In our work, the term VS can be used interchangeably with reward shaping and policy shaping. The distinction between reward shaping, VS, and policy shaping cannot be addressed in single-step problems such as in our task. In fact, adding an extra reward bonus to an action is equivalent to augmenting its expected value, which is equivalent to augmenting its probability of being selected. Further research using multistep reinforcement learning paradigms is needed to assess which of these 3 shaping methods best accounts for human behavior.

## Comparing value shaping to model-based imitation

Our results show that human Learners imitate an SD by integrating the observed actions into their own value function, without building an explicit model of the Demonstrator. This solution has the obvious advantage of being computationally simpler than MB. In our study, the distinction between VS and MB echoes the classical distinction between model-free and model-based methods in the reinforcement learning literature. At first view, our findings may seem in contrast with previously reported results showing that people infer a model of a Demonstrator [5,9] and can successfully predict others behavior [27]. These seemingly contradictory findings can be partially explained by the fact that in these works, participants were explicitly instructed to predict other participant's behavior; whereas in our tasks, they were not. So it might be that participants do not build a model of the Demonstrator unless there is a need for it or being explicitly asked for. Moreover, in these previous works, demonstrations were the only available source of information for the Learner, while in our task, demonstrations were in competition with self-generated outcomes. As the task is defined by the outcomes, they constitute a reference model over which other learning signals might be integrated. VS represents a more parsimonious solution than building separate models for every learning signal.

More generally, we can imagine that both learning methods exist and correspond to distinct and coexisting inference modes that are used in different contexts: When only demonstrations are provided, inferring the Demonstrator's goal is the only way for learning the task, whereas in presence of another source of information (i.e., self-generated rewards), demonstrations only play a secondary role.

Which of these learning modes is used for imitation can have distinct implications. MB implies that people build distinct representations of the same task, one of which can be a representation of a Demonstrator's goal. Each representation can then influence the others, be switched on and off, and be more or less considered by the Learner. This provides a certain control to the Learner on the capacity to disentangle the reliability of each model. VS, on the other hand, implies a deeper effect of imitation. People would integrate others' behavior and adopt their preferences as their own. This "subversive" effect of imitation can be found in other works [28], where over-imitation is explained not as a mimicry mechanism, but by the modification of the inner representation of the causal model of the environment.

## Adaptive modulation of imitation

Our findings are consistent with the repeated observations in theoretical studies indicating that imitation should be a controlled process [1,16,19]: For imitation to be adaptive, it should be modulated by several environmental and social signals. Here, we focused on the Demonstrator's skills and found that when the Demonstrator was not skilled, the imitation learning rate was down-regulated. In other terms, consistent with previous studies, we show that the adaptive regulation of imitation can be achieved by monitoring endogenous signals (likely an estimate of the Demonstrator's performance) and does not necessarily require explicit cues and instructions [7,8,14]. Interestingly, participants in Exp 1 were more successful in modulating their imitation of the UD compared to participants in Exp 2 (cf. Fig 3C). A possible interpretation could be that participants in Exp 1 had the opportunity to compare the performance of both type Demonstrators and thus avoid imitating the UD. In Exp 2, even though participants imitated the SD more than the UD, the effect was less strong than in Exp 1.

At the computational level, this effect can be formalized as a meta-learning process, where relevant variables are used to optimally tune the learning parameters [29,30]. For example, this could be achieved by tracking the difference between the reward earned following the Demonstrator's choices and opposite choices, then using this relative merit signal to adaptively modulate imitation [31]. However, in our task, imitation rates could not have been modulated by such hypothetical comparison of the outcomes because we did not present the Demonstrator's outcome. We proposed and tested an alternative modulation process that is based on the evaluation of the Demonstrator's choices in the light of the Learner current preferences. The basic intuition is that if the demonstrators' choices agree with what the learner would have done, the learner starts to "trust" the Demonstrator. Our model is inspired by several previous empirical findings showing that the human brain's reward network is responsive to agreement in decisions on matters of taste that have (by definition) no correct outcome [32,33]. Thus, the learner treats the Demonstrator's choice as a surrogate reward (VS) preferentially when they present an overall high agreement rate.

Consistent with this hypothesis, Boorman and colleagues [34] showed that in a market prediction task, learning of the expertise of another agent (the role akin to the Demonstrator in our task) was accelerated when he/she expressed a similar judgment. The meta-learning model managed to capture the adaptation of the imitation rate to the type of the Demonstrator across the 2 experiments. A caveat of our implementation is that it supposes that the initial imitation learning rate is 0, a simplifying assumption that may not reflect many real-life situations,

where imitation is the primary source of information. It should be noted that our meta-learning framework could easily be extended by assuming an additional free parameter determining the baseline imitation learning rate (we tested this implementation in our data, but it was rejected due to the additional free parameter). In Exp 3, we were able to find a significant modulation of imitation only when the participants were informed about the IQ of the Demonstrator. This finding indicates 2 things. First, for the case when the Demonstrator's IQ was hidden, the performance gap between the UD (approximately 0.65 correct choice rate) and the SD conditions (approximately 0.85) was not large enough to endogenously modulate imitation. Second, when the Demonstrator's IQ was visible, imitation could also be controlled by exogenous information about cognitive abilities. This finding is consistent with many studies showing that reputational priors shape learning and decision-making at the behavioral and neural levels [35,36].

Our model parameters analysis also showed that the imitation learning rate was smaller compared to the private reward learning rate, even when the Demonstrator outperformed the participant (80% average correct response rate versus <75%). This difference could derive from the fact that the Demonstrator's choice, as a proxy of reward, implies an additional degree of uncertainty (one will never know whether the Demonstrator obtained a reward, after all). This difference could also derive from an egocentric bias, where self-generated information is systematically overweighted. Whether or not the difference between reward and imitation learning rates changes as a function of task-related factors (e.g., task difficulty) or interindividual differences (e.g., age of the participants and pathological states) remains to be established [37–39].

## What about imitation in the action space?

In our experiments, the actual motor responses (i.e., the motor action) necessary to implement the decision of the Demonstrator were not shown. We communicated them in abstract terms. We opted for this implementation for three reasons: we wanted to stay as close as possible to original—and widely used—design [6]; we wanted to avoid implicit communication via body signals and movements; and we wanted to focus on pure imitation in the choice space (please note that an action observation protocol would also involve choice observation). However, our experiments leave unanswered whether the same results would hold in a situation where the Demonstrator's actions are observable. Concerning imitation in an action–observation context, we recognize two possible scenarios that will be addressed by future studies. In one scenario, VS requires processing the Demonstrator's behavior in the choice space and therefore imitation would revert to a DB process prompted by a motor contagion process [40]. In another scenario, imitation recycles the same VS computations in both the action and the choice space. Therefore, as the quality and quantity of social signals increases in the action–observation configuration, one could predict that the imitation learning rate could be as high as the private learning rate in these contexts (thus reverting the alleged egocentric bias).

## Putative neural bases

Our computational modeling results suggest that the actions of a Demonstrator constitute a pseudo-reward, generating a reward prediction error used for updating the Learner's value function. Previous imaging data provided evidence consistent with this neuro-computational hypothesis. Specifically, Lebreton and colleagues [41] showed, in a simple option valuation task, that action observation signals (originally encoded in the human mirror neuron system [42]) progressively affect reward-related signals in the ventral striatum and the ventral prefrontal cortex, two areas robustly associated with reward prediction error encoding in

reinforcement learning [43]. This neuro-computational hypothesis is also consistent with data showing action prediction errors in the lateral prefrontal cortex (overlapping with the mirror neuron system) and modulation of reward-related signals in the striatum and ventral prefrontal cortex, when following advice [6,11,44], in tasks where imitation relies on the communication of abstract social signals (e.g., choices and not actions).

## Conclusions

Our work provides robust evidence that, in the context of social reinforcement learning, imitation takes the form of a computationally simple, yet long-lasting, value shaping process, where the choices of the Demonstrator durably affect the value function of the Learner. Imitation plays a key role in the propagation of ideas and behaviors among individuals [45]. The global adoption of social media has exacerbated the propagation of different kinds of behaviors, spanning from anodyne memes and lifestyle habits, to more political [46] and economic [47] behaviors. These massive cascade effects can have deleterious impacts, such as political polarization [48], diffusion of fake news [49], and even more macabre trends caused by the propagation of dangerous behaviors [50]. Understanding the mechanisms underpinning social influence is therefore crucial for understanding and preventing these negative effects. Our main finding is that imitation can be understood as a modification of values rather than a bias over decisions. The ensuing persistence of imitation may in part explain the strength and pervasiveness of phenomena related to social influence. Future research will determine whether or not imitation is impaired in social and nonsocial psychiatric conditions, such as depression, anxiety, autism, and borderline personality disorder, and whether these hypothetical impairments take the form of shifts in its computational implementation and/or model parameters [51].

## Methods

### Ethics statement

The study was approved by the local Institutional Review Board (C15-98/58-17).

### Participants

Respectively 24 and 44—new—healthy participants were enrolled in our 2 experiments (Table 1). Participants were recruited through the *Relais d'Information sur les Sciences Cognitives* website (https://www.risc.cnrs.fr/). The inclusion criteria were being over 18 years old and reporting no history of psychiatric and neurological illness. All study procedures were consistent with the Declaration of Helsinki (1964, revised 2013), and participants gave their written informed consent, prior to the experiment. Participants were reimbursed 10 to 20 EUR for their participation, depending on their performance (on average 13.9 ± 4.69 EUR).

**Table 1. Demographics of the 2 cohorts of participants.**

|  | Exp 1 | Exp 2 | |
|---|---|---|---|
|  |  | SD | UD |
| # participants | 24 | 22 | 22 |
| # male–female | 10–14 | 9–13 | 11–11 |
| age mean (± sd) | 24.5 (±3.45) | 24.9 (±3.5) | 26.4 (±3.9) |
| # same-mixed gender | 9–15 | 9–13 | 10–12 |

# same-mixed gender, indicated whether the dyads were composed by 2 participants of the same or different genders. sd, standard deviation; SD, Skilled Demonstrator; UD, Unskilled Demonstrator.

The third experiment presented in this paper consists of previously published data. We refer to the original publication for full details about the participants [14].

## Experimental task and procedure

Both experiments implemented variants of a standard social reinforcement learning task, where, in addition to private learning trials and conditions, the participants were sometimes exposed to the choices of another agent. The design was based on a previous observational reinforcement learning task [6], where we changed several features. First, as we were interested in pure imitation (and not vicarious trial-and-error learning), we did not include observation of the Demonstrator's outcomes. Second, to assess both the accumulation and the propagation of the Demonstrator's influence, we allowed for more than 1 consecutive observational or private learning trial (while keeping their total number equal). In fact, over both experiments, the number of demonstrations in a row varied between 1 and 7 demonstrations, with the additional constraint of not having more than 3 consecutive demonstrations of the same option. Third, we manipulated the Demonstrator's performance by implementing skilled (80% of optimal, i.e., reward maximizing responses) and unskilled (20% of optimal responses) behavior. This allowed us to assess whether imitation was adaptively modulated. We did not implement unskilled performance as 50% of optimal responses, as it would have confounded 2 factors: optimality and variability (indeed a 50% correct Demonstrator would switch options continuously, while a 20% correct Demonstrator is suboptimal but as stable as the skilled one).

Other details concerning the experimental task are provided in the Results section. Participants were recruited in pairs—either mixed or same gender—and received the instructions jointly. Participants were tested in 2 separated cabins and took breaks at the same time. They were asked in advance to provide an electronic version of a passport-like photo (displayed on the upper part of each side of the computer screen) to clearly differentiate private and observational trials. They were told they would engage in a small game in which they could sometimes observe each other's choice in real time—but not the outcomes associated with that action. Participants were informed that some symbols would result in winning more often than others and were encouraged to accumulate as many points as possible. It was emphasized that they did not play for competition but for themselves, and importantly, there was no explicit directive to observe the other player. The participants were not explicitly instructed to integrate the choices of the other player into their own learning and decision-making, but it was explicit that a given symbol had the same value for both participants. At the end of each session, the experimenter came in both cabins to set the subsequent one, and participants were instructed to begin each session at the same time. The debriefing of the experiment was done with each pair of participants, and all participants were debriefed about the cover story after the experiment. The third dataset presented in this paper comes from a previously published study [14]. The main differences between our design and theirs are summarized in Table 2 and include

**Table 2. Main differences between experiments.**

|                              | Exp 1            | Exp 2            | Exp 3                          |
|------------------------------|------------------|------------------|--------------------------------|
| # participants               | 24               | 44               | 302                            |
| # trials                     | 400              | 300              | 300                            |
| Observational:private trials | 1:1              | 1:1              | 1:2                            |
| Contingencies                | Stable (70/30)   | Stable (60/40)   | Random walk                    |
| Demonstrators                | Computer         | Computer         | Real participants              |
| Demonstrator skills          | 0.80 / 0.20      | 0.80 / 0.20      | $0.84 \pm 0.03$ / $0.60 \pm 0.05$ |
|                              | (within)         | (between)        | (between)                      |

The Demonstrator's skills are defined by the correct choice rate. The variability on the Demonstrator's skills of Exp 3 is given in standard errors of the mean.

reward contingencies changing on a trial-by-trial basis (random walk); the Demonstrator choices consisted in recorded choices of previous participants playing exactly the same contingencies with no social component (therefore in this experiment no deception was involved); the ratio between observational and private trials was 1:2 (and not necessarily the same demonstrations were presented to the participants); some participants (roughly 50%) received information about the IQ (raven matrices) of the Demonstrator. We refer the readers to the original publication for full details of the experimental task [14].

## Computational modeling

We compared three hypotheses about the computational implementation of imitation in reinforcement learning. All 3 models are the same for learning from self-experienced outcomes. They only differ in how they integrate demonstrations into the learning process.

## Baseline model

As a baseline, we implemented an RW model that learns only from self-experienced outcomes while ignoring demonstrations. A value function, Q, is first initialized to 0. Then, on every time step, when the participant chooses the option $c$ and observes an outcome $r$, the value of the chosen option is updated as follows:

$$Q(c) \leftarrow Q(c) + \alpha_p \times [r - Q(c)], \tag{2}$$

where $\alpha_p$ is a "private" reward learning rate. Action selection is performed by transforming action values into a probability distribution through a softmax function

$$\pi(c) = \frac{1}{1 + e^{\beta \times [Q(\bar{c}) - Q(c) - \lambda(c).\kappa]}}, \tag{3}$$

where $Q(\bar{c})$ is the value of the unchosen option $\bar{c}$, $\kappa$ is a choice autocorrelation parameter [52], and $\lambda(c)$ is defined as

$$\lambda(c) = \begin{cases} 1 & \text{if c is the last performed action} \\ -1 & \text{otherwise} \end{cases}$$

We compared 2 implementations of this baseline depending on how the Q-values are updated. The first implementation, RW1, uses the value update scheme described in Eq 2. The second implementation, RW2, is based on a symmetric value update that also updates the value of the unchosen option in the opposite direction with

$$Q(\bar{c}) \leftarrow Q(\bar{c}) + \alpha_p \times [-r - Q(\bar{c})].$$

## Decision biasing

DB builds on the baseline when it comes to learning from experienced outcomes. However, demonstrations are also used for biasing action selection. We tested 6 different implementations of this model in order to control for some computational details that may affect the fitting quality (cf. Fig 7). The first implementation, DB1, is the original model presented in [6] and used in [7]. When a demonstration $d$ is observed, it is used for biasing the policy $\pi$ as follows. First, the policy $\pi$ is derived from the value function Q via Eq 3. Then, the policy $\pi$ is modified via

$$\pi(d) \leftarrow \pi(d) + \alpha_i \times [1 - \pi(d)],$$

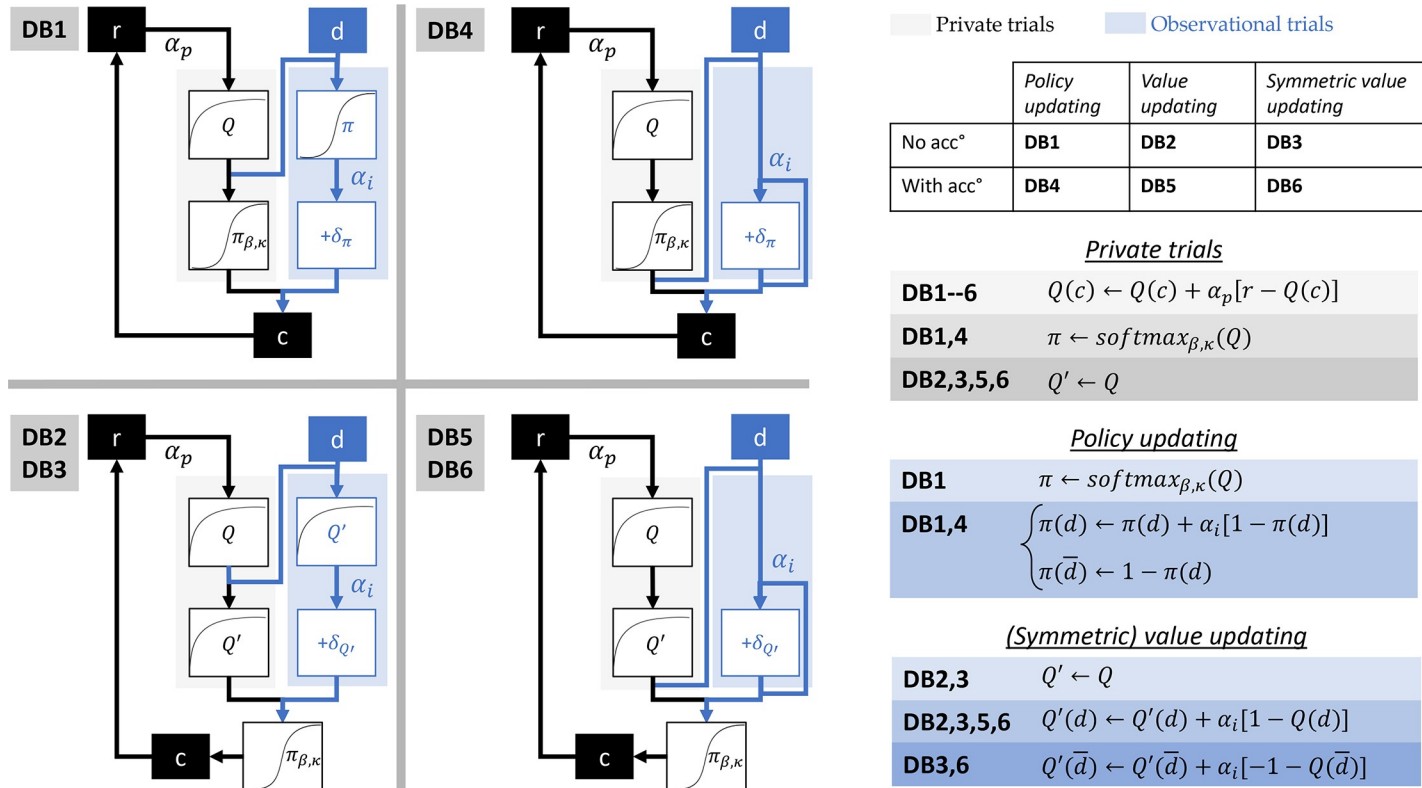

**Fig 7. Six implementations of DB.** In DB1, demonstrations bias Learner's actions via policy update. DB2 and DB3 implement the same mechanism through value update and symmetric value update. DB4, DB5, and DB6 are equivalent to respectively DB1, DB2, and DB3, while allowing for the accumulation of successive demonstrations. This is done by removing the first step of the update where $\pi$ or $Q'$ is derived from $Q$. Accumulation is depicted in the diagrams by the loop within observational trials. DB, decision biasing.

$$\pi(\bar{d}) \leftarrow 1 - \pi(d), \tag{4}$$

where $\pi(d)$ and $\pi(\bar{d})$ represent, respectively, the probability to select the demonstrated and the non-demonstrated option, and $\alpha_i$ an imitation decision bias rate parameter.

We refer to this update scheme as policy update. In order to control for the fact that in DB the action prediction error is computed at the level of the policy and not the Q-values, we devised another version, DB2, in which the action prediction error is computed based on the Q-value (i.e., a value update). But still, this update must only affect action selection without modifying the value function. To do so, we keep a "decision value function," $Q'$, which is a copy of the value function $Q$ that is used only for action selection. The policy $\pi$ is derived from $Q'$ instead of $Q$. When a demonstration $d$ is observed, it is used for biasing $Q'$ as follows. First, $Q'$ is copied from $Q$. Then, it is modified via

$$Q'(d) \leftarrow Q'(d) + \alpha_i \times [1 - Q'(d)]. \tag{5}$$

One difference between Eq 4 and Eq 5 lies in the symmetry of the update. In Eq 4, increasing the probability of selecting one option naturally decreases the probability of selecting the other option. However, in Eq 5, increasing the value of one option does not affect the value of the alternative option. To account for this fact, we implemented an extension of DB2, in which we also update the value of the alternative option in the opposite direction. DB3 implements a

symmetric value update via

$$Q'(d) \leftarrow Q'(d) + \alpha_i \times [1 - Q'(d)],$$

$$Q'(\bar{d}) \leftarrow Q'(\bar{d}) + \alpha_i \times [-1 - Q'(\bar{d})]. \tag{6}$$

One common limitation of all these 3 implementations is that they do not allow the accumulation of successive demonstrations, because the policy $\pi$ (resp. $Q'$) is derived from $Q$ each time a demonstration is provided. To address this limitation, we extend these implementations, by removing the initial step of the update that derives $\pi$ (resp. $Q'$) from $Q$. This way, successive demonstrations within the same block accumulate their effects.

## Model-based imitation

Just like DB, MB is built on the baseline for learning from self-experienced outcomes. However, demonstrations are used for building a model of the demonstrator's preferences. This model is then used for biasing action selection. We compared 9 different algorithmic implementations of this model (cf. Fig 8). These implementations differ in how the model of the Demonstrator is built and how it is used for biasing action selection. We construct a 3×3 model space based on whether each step uses a policy update (Eq 4), a value update (Eq 5), or a symmetric value update (Eq 6). To note, a symmetric value update of the Demonstrator's model corresponds to the approximate inverse reinforcement learning algorithm implemented in [9]. In all 9 implementations, the Demonstrator's model is updated with a learning rate $\alpha_d$ and used for biasing action selection with an imitation decision bias $\alpha_i$. Finally, for each of these 9 implementations, we compared 2 versions, one where $\alpha_d$ is fitted as a free parameter, and one where it is fixed to 0.1 (cf. S5 Fig).

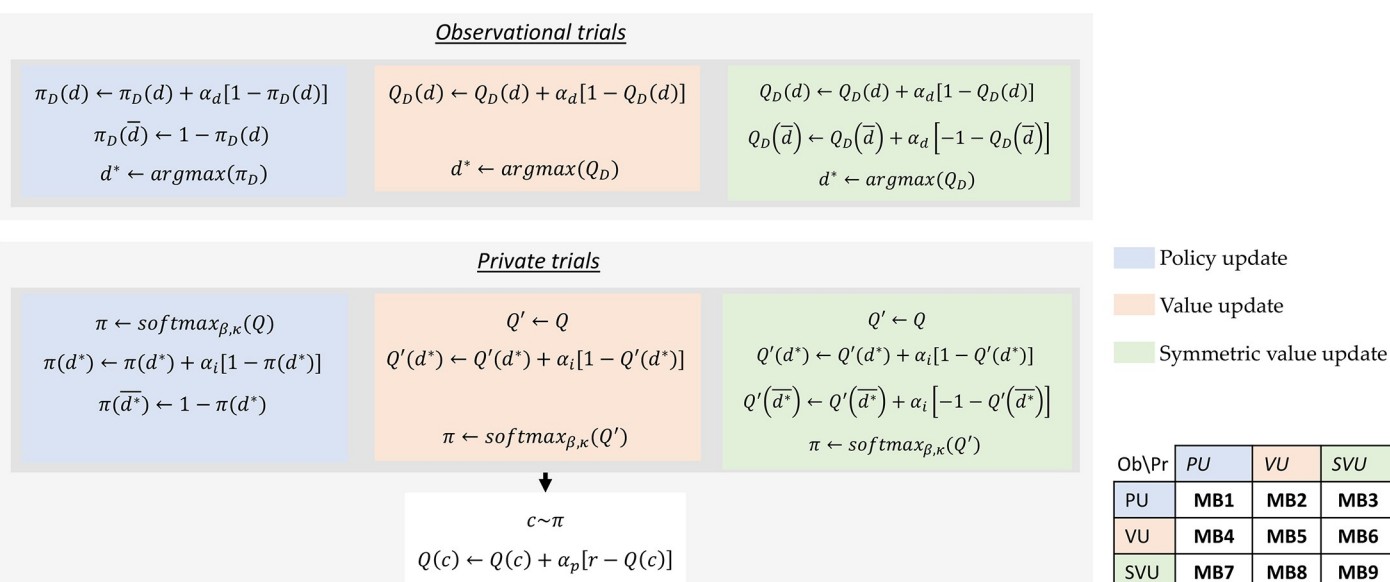

**Fig 8. Nine implementations of MB.** In observation trials, demonstrations are used for building a model of the Demonstrator ($\pi_D$ or $Q_D$). This is done either through policy update (MB1, MB2, and MB3), value update (MB4, MB5, and MB6), or symmetric value update (MB7, MB8, and MB9). In private trials, the model of the Demonstrator is used for biasing Learner's actions through either policy update (MB1, MB4, and MB7), value update (MB2, MB5, and MB8), or symmetric value update (MB3, MB6, and MB9). MB, model-based imitation; PU, policy update; SVU, symmetric value update; VU, value update.

## Value shaping

In VS, demonstrations are directly integrated in the Learner's value function by reinforcing the demonstrated actions. We compared 2 implementations of this model. VS1 performs a value update via

$$Q(d) \leftarrow Q(d) + \alpha_i \times [1 - Q(d)] \tag{7}$$

VS2 performs a symmetric value update by also updating the value of the non-demonstrated option via

$$Q(\bar{d}) \leftarrow Q(\bar{d}) + \alpha_i \times [-1 - Q(\bar{d})], \tag{8}$$

## Meta-learning

To account for the dynamic modulation of the imitation learning rate, we designed a meta-learning model. The core idea of the model is that the skill of the Demonstrator is inferred by comparing her choices to the current knowledge of the Learner. The imitation learning rate $\alpha_i$ in the meta-learning model (meta-VS) is first initialized at 0 for every state (i.e., pair of cues). Then, $\alpha_i$ is dynamically modulated on a trial-by-trial basis depending on whether or not the current observed action maximizes the Learner's Q-values. The modulation is performed using an auxiliary learning rate $\alpha_m$ that measures how fast a participant learns about the approximate skill of the Demonstrator:

$$\alpha_i(s) \leftarrow \alpha_i(s) + \alpha_m * (\tau - \alpha_i(s)),$$

where $s$ represents the current state (pair of cues), and

$$\tau = \begin{cases} 1 & \text{if } Q(d) = max(Q(d), Q(\bar{d})) \\ 0 & \text{otherwise} \end{cases}$$

Note that, in meta-VS, $\alpha_m$ is a free parameter, whereas $\alpha_i$ is not. The imitation rate is then used for updating the Learner's value function through value-shaping using Eqs 7 and 8.

## Model fitting procedure

**Model comparison and parameter optimization.**   In a first phase, we compared the different implementations of each model. The best implementation was then included in the final model space for comparison (cf. S2 Fig).

Each model was fitted separately to the behavioral data of our 3 experiments. First, the free parameters of each model were optimized as to maximize the likelihood of the experimental data given the model. To do so, we minimize the negative log-likelihood using the *fmincon* function in Matlab, while constraining the free parameters within predefined ranges ($0<\alpha<1,0<\beta<+\infty$). The negative log-likelihood NLL is defined as

$$NLL = -log(P(Data|Parameters, Model)).$$

Then, from the negative log-likelihood we derive the Akaike Information Criterion (AIC) [53] defined as

$$AIC = -NLL - p,$$

where $p$ is the number of free parameters.

The AIC is—with other metrics—a commonly used metric for estimating the quality of fit of a model while accounting for its complexity. As such, it provides an approximation of the out-of-sample prediction performance [54]. We selected the AIC as a model comparison metric after comparing its performance in model recovery along with other metrics (see below).

Finally, the individual AIC scores were fed into the mbb-vb-toolbox as an approximation of model log-evidence [55]. Contrary to fixed-effect analyses that average the criteria for each model, the random-effect model selection allows the investigation of interindividual differences and to discard the hypothesis of the pooled evidence to be biased or driven by some individuals—i.e., outliers [56]. This procedure estimates the model expected frequencies and the exceedance probability for each model within a set of models, given the model log-evidence for each participant. The expected frequency is the probability of the model to generate the data obtained from any randomly selected participant—it is a quantification of the posterior probability of the model (PP). It must be compared to chance level, which is one over the number of models in the model space. The exceedance probability (XP) is the probability that a given model fits the data better than all other models in the model space. Theoretically, a model with the highest expected frequency and the highest exceedance probability is considered as "the winning model."

Model parameters were estimated by maximizing the Log Posterior Probability (LPP):

$$LPP = log(P(Parameters|Data, Model))$$

LPP maximization includes priors over the parameters (inverse temperature gamma (1.2,5); LR beta (1.1.,1.1). Essentially, it avoids wrongful fitting of the parameters estimates that could be driven by noise. The same priors were used for all learning rates to avoid bias in learning rate comparison. Of note, the priors are based on previous literature [70] and have not been chosen for this study. While the LPP, in principle, could be used to compare models (as it integrates the probability of the parameters, therefore penalizing a higher number of parameters), it is usually not a very stringent criterion (especially when the parameter priors are weakly informative). Accordingly, in our case, it did not give a very good model recovery (cf. S9 Fig).

## Model recovery

The main aim of the model recovery procedure is to verify that the models that we try to compare can effectively be distinguished given the experimental data and the model fitting procedure. It could be the case that 2 competing models are not sufficiently different from each other to be effectively distinguished or that the experimental design is not appropriate for eliciting any difference between these models.

Model recovery consists in running several simulations of each model with the historical data of each participant, i.e., the observed stimuli and demonstrations. Model parameters are initialized randomly according to their respective prior distributions. This allows us to have a ground truth about which model has generated each piece of data. After simulation, we run the model fitting procedure on the data generated by each model and report the model frequency and exceedance probability of each competing model. If the exceedance probability of the model that has generated the data passes a predefined threshold, it means that our model fitting procedure allows us to recover the true generating model. Our model recovery analysis confirmed that, in our task and for our models, the AIC is a good criterion, while LPP is not.

## Parameter recovery and comparison between parameters

Parameter recovery is a useful way to assess the quality of the parameter fitting procedure. To do this, we computed the Spearman correlation between parameter values that were used for

generating simulated data, with the values recovered by the model fitting procedure. A high correlation score indicates a reliable parameter fitting procedure. The good parameter discovery displayed by the winning model indicated that their estimates are meaningful and can be statistically compared. As assumptions about normal distributions were not verified, we used nonparametric tests for comparing model parameters. Wilcoxon signed-rank test was used for comparing paired learning rates, such as private versus imitation learning rates (Fig 3C) and skilled versus unskilled imitation learning rates in Exp 1 (Fig 3C). Wilcoxon rank-sum test was used for non-paired learning rates, such as skilled versus unskilled imitation learning rates in Exp 2 (Fig 3C).

## Supporting information

**S1 Fig. Raw task performance on both experiments, with respect to conditions.** P private condition. SD and UD observational conditions. In Exp 1, task performance in the SD condition was higher than in the Private condition (P), which itself was higher than performance in the UD condition. However, this difference in performance was not statistically significant. In Exp 2, the differential in task performance between observational and private conditions was statistically different between the SD and the UD group. Underlying data can be found in https://github.com/hrl-team/mfree_imitation/. SD, Skilled Demonstrator; UD, Unskilled Demonstrator.
(TIF)

**S2 Fig. Intermediate model comparisons.** We compared different implementations of each model: 2 for RW, 6 for DB, 9 for MB, and 2 for VS. For the baseline, implementing a symmetric value update (RW2) improves the model fitting quality in Exp 1. For DB, the best fitting performance is achieved when allowing for symmetric value update from observed demonstrations and for the accumulation of successive demonstrations (DB6). For MB, the best implementations use the model of the Demonstrator for biasing Learner's actions through symmetric value update (MB3 and MB9 in Exp 1 and MB6 in Exp 2). A finer comparison between MB3 and MB9 in both experiments shows that these models are equivalent (not shown here). A further comparison between MB6 and MB9 in both experiments shows that MB9 fits better than MB6. Finally, the best VS implementation, VS2, uses a symmetric value update from observed demonstrations. As a result of this analysis, the final model space includes RW2, DB6, MB9, and VS2. Underlying data can be found in https://github.com/hrl-team/mfree_imitation/. DB, decision biasing; MB, model-based imitation; RW, Rescorla–Wagner; VS, value shaping.
(TIF)

**S3 Fig. Model comparison results are robust among different implementations of the model space.** (a) Model space implementation without choice autocorrelation parameter. (b) Model space implementation without symmetric value update for private learning. (c) Model space implementation allowing for negative imitation learning rates. Note that in Exp 2, when allowing for negative learning rates, the winning model in the UD condition is no longer RW, but VS. Underlying data can be found in https://github.com/hrl-team/mfree_imitation/. RW, Rescorla–Wagner; UD, Unskilled Demonstrator; VS, value shaping.
(TIF)

**S4 Fig. Parameter comparison between UD and SD conditions.** No statistical difference was found for nonsocial parameters $\alpha_p$, $\kappa$, and $\beta$. Only the imitation learning rate $\alpha_i$ was statistically different across observational conditions in both experiments. Underlying data can be found in https://github.com/hrl-team/mfree_imitation/. SD, Skilled Demonstrator; UD,

Unskilled Demonstrator.
(TIF)

**S5 Fig. Model comparison between alternative implementations of MB.** Specifically, implementations with a fixed parameter $\alpha_d = 0.1$ (MB3, MB6, and MB9) are better than implementations with $\alpha_d$ as a free parameter (MB3+, MB6+, and MB9+, respectively). $\alpha_d$ is the learning rate used for inferring the preferences of the Demonstrator. Underlying data can be found in https://github.com/hrl-team/mfree_imitation/. MB, model-based imitation.
(TIF)

**S6 Fig. Model properties.** Accumulation (top row): behavioral imitation rate calculated as a function of the 2 preceding demonstrations ($d_{t-1}$, $d_{t-2}$). Propagation (bottom row): behavioral imitation rate calculated as a function of 2 consecutive choices ($c_{t+1}$, $c_{t+2}$). $^*p<0.05$, $^{**}p<0.01$, $^{***}p<0.001$, paired $t$ test. Underlying data can be found in https://github.com/hrl-team/mfree_imitation/.
(TIF)

**S7 Fig. Model predictions.** Across-trial correlation between the observed and model-predicted choices. Results are given as Spearman's correlations. Underlying data can be found in https://github.com/hrl-team/mfree_imitation/.
(TIF)

**S8 Fig. Model predictions.** Across-subject correlation between the observed and model-predicted choices. Results are given as Spearman's correlations. Underlying data can be found in https://github.com/hrl-team/mfree_imitation/.
(TIF)

**S9 Fig. Model recovery using different approximations of model log-evidence for the main computational models.** We tested different approximations of the model evidence that we fed to the Variational Bayesian Analysis toolbox. (a) AIC. (b) log-likelihood. (c) log posterior probability. Only AIC displayed a good parameter recovery. AIC, Akaike Information Criterion.
(TIF)

**S10 Fig. Model comparison using the likelihood as approximation of the model evidence and restricting the comparison to the 3 imitation models.** We compared the main imitation models (DB, MB, and VS) with maximum likelihood because they present the same number of free parameters. Underlying data can be found in https://github.com/hrl-team/mfree_imitation/.
(TIF)

## Author Contributions

**Conceptualization:** Anis Najar, Bahador Bahrami, Stefano Palminteri.

**Data curation:** Anis Najar, Emmanuelle Bonnet.

**Formal analysis:** Anis Najar.

**Funding acquisition:** Stefano Palminteri.

**Investigation:** Emmanuelle Bonnet.

**Methodology:** Anis Najar, Emmanuelle Bonnet, Stefano Palminteri.

**Project administration:** Stefano Palminteri.

**Supervision:** Stefano Palminteri.

**Validation:** Stefano Palminteri.

**Visualization:** Anis Najar, Stefano Palminteri.

**Writing – original draft:** Anis Najar, Stefano Palminteri.

**Writing – review & editing:** Anis Najar, Emmanuelle Bonnet, Bahador Bahrami, Stefano Palminteri.

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
