## [Editor Report · Decision Letter 0]

3 Nov 2019

Dear Dr Palminteri, 

Thank you for submitting your manuscript entitled "Imitation as a model-free process in human reinforcement learning" for consideration as a by PLOS Biology.

Your manuscript has now been evaluated by the PLOS Biology editorial staff, as well as by an Academic Editor with relevant expertise, and I am writing to let you know that we would like to send your submission out for external peer review. Please accept my apologies for the delay in sending this decision to you.

Please re-submit your manuscript within two working days, i.e. by Nov 06 2019 11:59PM.

Kind regards,

Gabriel Gasque, Ph.D.,

Senior Editor

PLOS Biology

---

## [Decision Letter · Decision Letter 1]

9 Dec 2019

Dear Dr Palminteri,

Thank you very much for submitting your manuscript "Imitation as a model-free process in human reinforcement learning" for consideration as a Research Article at PLOS Biology. Your manuscript has been evaluated by the PLOS Biology editors, by an Academic Editor with relevant expertise, and by three independent reviewers.

Based on reviewers' specific comments and following discussion with the Academic Editor, I regret that we cannot accept the current version of the manuscript for publication. We remain interested in your study and we would be willing to consider resubmission of a comprehensively revised version that thoroughly addresses all the reviewers' comments. We cannot make any decision about publication until we have seen the revised manuscript and your response to the reviewers' comments. Your revised manuscript would be sent for further evaluation by the reviewers.

The reviews of your manuscript are appended below. You will see that the referees find the manuscript mostly well written, timely, and technically solid. However, they also concur that, at this stage, the findings are specialized, and that additional data and analyses are required for providing a more general scope and appeal, as well as to solidify your conclusions. The reviewers have raised several points, many of which can be addressed by clarification and rewriting and, in some cases, by further analysis. However, some of the reviewers’ points involve requests for additional data. Having discussed these requests for additional data with the Academic Editor, we think that the key issues that might need additional data are as follows:

Reviewer 2’s point concerning “adaptive modulation of imitation” and reviewer 3’s related concern summarized in their major point 2. In both cases, the reviewers argue that additional data may be needed to make a definitive conclusion and reviewer 2 argues that additional analyses are needed too. 

Reviewer 2’s point concerning “Reward shaping/policy shaping/value shaping” and reviewer 3’s related concern summarized in their major point 1. Again, both reviewers suggest that ideally additional data are needed to address these concerns. 

We appreciate that these requests represent a great deal of extra work, and we are willing to relax our standard revision time to allow you six months to revise your manuscript.We expect to receive your revised manuscript within 6 months.

**IMPORTANT - SUBMITTING YOUR REVISION**

*NOTE: In your point by point response to to the reviewers, please provide the full context of each review. Do not selectively quote paragraphs or sentences to reply to. The entire set of reviewer comments should be present in full and each specific point should be responded to individually, point by point.

*Resubmission Checklist*

*Published Peer Review*

*PLOS Data Policy*

*Blot and Gel Data Policy*

Sincerely,

Gabriel Gasque, Ph.D., 

Senior Editor

PLOS Biology

REVIEWS:

Reviewer #1: Najar et al. have examined how observing others' actions affect one's own value-based decision-making in the context of instrumental learning. Fitting various computational models to the behavioral data and comparing their goodness of fit, they have demonstrated that observation of others’ actions directly affects the observer’s value function in a computationally simple way. They have further shown that the imitation process is modulated by how much the others' actions are informative (i.e., expertise or skill-level). I believe the authors have addressed an important issue in human social decision-making with a solid experimental design and careful data analyses. Their findings would therefore provide significant insights into psychological mechanisms underlying human social decision-making. On the other hand, I am not sure whether the present study could be of wide interest not only to researchers who are interested in human social decision-making.

The authors have demonstrated that one class of the models (i.e., value-shaping) outperformed the other classes (i.e., decision-bias and model-based imitation). What is the qualitative difference among the three types of the models? In other words, what types of behavioral pattern in this experiment and the real world can be "exclusively" explained by the winning model? To answer this question by model-based and/or model-free analyses might increase the novelty and significance of the study.

The data indicate that participants infer the expertise of a Demonstrator. It would be interesting to construct computational models that incorporate learning about others' expertise. As far as I know, it remains elusive whether and how learning from others' action (e.g., Burke et al., 2010 and the present study) is integrated with learning about others' expertise (e.g., Boorman et al., 2013). Examination of this issue could reinforce the novelty and significance of the study. In relation to this issue, I don't agree with the authors' claim that participants do not build a model of the Demonstrator. They clearly have a model of the Demonstrator, and it modulates the participants' imitation behavior.

It is difficult for me to understand details of the computational models. The authors should provide full descriptions of the models. Short descriptions combined with schematic diagrams are not sufficient to reproduce the models in future studies. Having said that, I would like to suggest other possible models worth tested.

[1] Baseline model. In the current model, $lambda$ reflects only the last action. Another way is to assume accumulated choice-trace affects decision-making (e.g., Akaishi et al., Neuron, 2014). What happen if the latter formulation is employed? Note that the latter formulation of a choice auto-correlation include the former original formulation as a special case.

[2] Decision biasing models. How about a model, in which participants keep track of the Demonstrator’s accumulated choice-trace (rather than the current action $d$) and the choice-trace works on the action-selection process?

[3] Model-based imitation models. How about a model, in which model-based inference about the Demonstrator’s value function affects the participant’s value function (rather than decision-making)?

I am not fully convinced by the authors' claim about egocentric bias. They have compared learning rate for participants' own REWARD and that for the Demonstrator's ACTION. The differential learning rates observed in this study can be attributed into the difference between learning from REWARD and from ACTION (rather than difference between self and other).

I would like to make sure the model-fitting procedures in this study. As far as I understand, for the model comparison the authors fit each model to the behavioral data by minimizing negative log likelihood (i.e., maximum likelihood estimation), while the parameter estimation is performed based on log model evidence (derived by Laplace approximation). Why did they use different ways for model comparison and parameter estimation? This looks weird. Furthermore, P(data | model) is "model evidence", not "maximum likelihood". Moreover I believe AIC cannot be fed into Bayesian model comparison. The authors should double-check.

Reviewer #2: Review: Imitation as a model-free process in human reinforcement learning

In this manuscript, the authors aim to elucidate the computational underpinnings of imitation, defined as using another’s actions a source of information in a probabilistic two-armed bandit task, in human reinforcement learning. The authors tackle an important question in a methodologically rigorous manner. As such, I find the manuscript interesting, timely, and well written. However, given the scope of the question (“the exact computational implementation of how social signals/imitation influence human reinforcement learning”, paraphrased from the abstract), and the quality level of the journal, it is my opinion that both additional data and analysis are required for providing general and conclusive answers to the research question. In short, I think the authors should be more ambitious in answering open questions they themselves identify. I outline my concerns and suggestions below, beginning with what I consider more major points.

Reward shaping/policy shaping/value shaping: The authors appropriately acknowledge (page 7) that the single-step experiment does not allow distinguishing whether imitation shapes value, reward, or policy, which instead would require a multi-step design. Given that the goal of the study was to determine the algorithmic implementation of imitation, this remains an important open question. I don’t understand why the authors did not attempt to address this question by also running a suitable multi-step experiment. Aside from being novel on its own right (I am not familiar with any multi-step designs in human social RL), such an experiment would allow addressing the important open question, and (possibly) confirm that the results generalize outside of the confines of the classic Burke (2010) design. 

Adaptive modulation of imitation: The authors find, by means of separately fitting the skilled (SD) vs unskilled demonstrator (UD), conditions that people are able to adjust their level of imitation (i.e., people don’t imitate in the UD conditions, as shown by the fit of the RW model). Furthermore, the authors discuss (pages 8-9) this in terms of a meta-learning process, and offer some suggestions for what information participants might use. Despite these observations, they do not, to my surprise, test these suggestions with additional modeling. I am therefor left rather unsatisfied with the modeling analysis, which is somewhat coarse (in fitting different within-participant conditions with different models). To me, a convincing account of the mechanisms underlying human imitation in RL would specify mechanisms that determine when people imitate (and thereby fit both SD and UD with the same model in Exp. 1). I also assume that this would be the goal of the experimental design (why else would both SD and UD conditions be included?).

On a related note, I do not understand why the UD condition has 20% optimal demonstrator choices, given that this means that the correlation between demonstrator choices and reward would be the same in both the UD and SD conditions, but with different signs. This should be clarified. Ideally, the authors would confirm in an additional control experiment that the same conclusions hold if the demonstrators choices are uncorrelated with reward (50% optimal choices), or at minimum, use simulation to determine expectations. 

* It is not completely clear whether the outcomes of the demonstrator was shown (which typically is the case in certain conditions in this experimental paradigm). I concluded that they were not, given the focus on imitation and model description. Nonetheless, this could be clarified.

*I find the description of the model fitting procedure confusing. Under the sub header “Model comparison”, its stated that models were fit using standard maximum likelihood, while under sub header “Parameter optimization”, its stated that a Bayesian technique (LPP) is used. I do not follow how model estimation and parameter estimation are two different things, given that the parameter values are estimated in the maximum likelihood fit of the model to the data. This should be clarified.

*The authors repeatedly state that privately generated outcomes are over weighed compared to social information (“egocentric bias”). It is unclear how this conclusion is reached, which should be clarified throughout. Furthermore, little theoretical basis for the importance of an egocentric bias is provided.

*The authors should plot average trial by trial data, together with generative model predictions. In the current presentation, the reader cannot know how well the preferred model fits the data.

*It would be informative to include generative model simulations of the various hypotheses to determine the adaptive value of each postulated mechanism. This would put the various hypothesis on firmer theoretical ground. 

*I understood that the experiments varied the number of demonstrator choices shown before the individual choice. However, I could not find any information on how this was implemented (e.g., how often, how many), and if the varying amount of social information differentially affected behavior.

Reviewer #3: This paper applies reinforcement learning models to the data of two experiments to investigate how imitation influences decision making. The results reveal that this influence can be best understood as “value shaping”, meaning that observers change their valuation of a choice after seeing someone else choose it. By doing so, the current paper provides insight into the mechanisms underlying social learning. 

Before starting my review, I would like to note that I am not a modeler. Therefore, although I am familiar with reinforcement learning models and understand the authors’ approach and results, I cannot comment on the specific implementation of the models.

I think this is an interesting paper with potentially important implications for how we understand social learning, and in particular for understanding why social influences can sometimes be very pervasive. That said, I also think the authors’ operationalization of imitation has important restrictions and disagree with their operationalization of demonstrator skill. This is important, as it is not clear at this point to what extent these operationalizations influence the modeling results. Solving these issues will, in my opinion, require additional experiments. Please find a detailed overview of my comments below.

Major comments

1. In the presented experiments, imitation takes the form of seeing the output of someone’s actions (i.e., their choice) and then deciding to copy that choice. One problem with this approach is that observers don’t actually see the demonstrator’s actions. That is, we know from research on automatic imitation that humans tend to imitate other people’s actions and that this tendency relies on motor contagion (Heyes, 2011, Psych Bull). Presumably, in the RL models used here, motor contagion would take the form of “decision biasing”. I therefore wonder if different models would be preferred if observers saw not only the outcome of the demonstrator’s actions but also the actions themselves. Perhaps, in this case, a hybrid model combining decision biasing and value shaping would best explain the results? Addressing this issue will require either additional experiments or a careful discussion of the limitations of the current approach.

2. The manipulation of demonstrator skill is not a manipulation of skill. In the experiments, the “unskilled” demonstrator chooses the “suboptimal” option in 80% of the trials. A true unskilled demonstrator, on the other hand, would pick the suboptimal option in 50% of the trials. The fact that the “unskilled” demonstrator chooses the suboptimal choice more often implies that this demonstrator has different preferences (i.e., for some reason they prefer to get less reward). This is not trivial: a true unskilled demonstrator is a demonstrator that contains no information and should therefore be ignored to optimize reward. An unskilled demonstrator as implemented here, on the other hand, is a demonstrator that should be counter-imitated to optimize reward. It is currently unclear to what extent this can explain the modeling results: is the value shaping mechanism a general mechanism or is it a mechanism that only applies to this specific situation? To address this question, I think additional experiments are needed that test whether the behavioral results differ depending on the operationalization of “skill” and whether the model is able to capture this.

3. Related to my first point, is it possible that hybrid models (e.g., a model combining model-based imitation and value shaping) would explain the results even better than a model with value shaping alone? Please discuss.

4. P. 14: “We selected the AIC as a metric after comparing its performance in model recovery along with other metrics such as the BIC (see the paragraph about model recovery)”. Why was the AIC chosen over the BIC? The AIC is known to sometimes overfit. Do the results change if the BIC is used instead? Also, I could not find anything on the BIC in the model recovery paragraph in contrast to what is suggested in the cited sentence. Please fix or clarify.

Minor comments

1. Did the authors check whether participants believed they were actually playing against someone else? In my experience, participants tend to be skeptical of such cover stories.

2. P 7.: “Over two experiments, we found that whenever imitation is adaptive (i.e., Skilled Demonstrator), it takes the computational form of value shaping, which implies that the 185 choices of the Demonstrator affect the value function of the Learner.”  Presumably, value shaping is always happening, but can be top-down modulated to have more or less effect on behavior. This is also how I interpret the “Adaptive modulation of imitation” section in the discussion. The cited sentence, on the other hand, makes it appear as if value shaping only happens for skilled demonstrators. Please change or clarify.

---

## [Decision Letter · Decision Letter 2]

27 Aug 2020

Dear Dr Palminteri,

Thank you for submitting your revised Research Article entitled "Flexible imitation as a model-free process in human social reinforcement learning" for publication in PLOS Biology. I have now obtained advice from the original reviewers 1 and 3 and have discussed their comments with the Academic Editor. The Academic Editor also assessed how you responded to reviewer 2's original concerns, since this reviewer couldn't re-review. 

We're delighted to let you know that we're now editorially satisfied with your manuscript. However, we would like you to consider changing your title to make it more accesible to a broad readerships and to convey the central biological message. We offer you this suggestion, but would be happy to discuss alternatives:

"The actions of others act as a vicarious reward in the context of reinforcement learning."

Please also check if the word "imitation" was not mistakenly inserted in the following sentence in the Introduction: “We also analyzed the parameters of the best winning model to determine whether or not, in the context of reinforcement learning, imitation more weight is given to information derived from oneself compared to the other [13].”

Also consider changing "her" for "their" in the following sentence in the Abstract: "The first hypothesis, decision biasing, postulates that imitation consists in transiently biasing the learner's action selection without affecting HER value function. "

In addition, before we can formally accept your paper and consider it "in press", we also need to ensure that your article conforms to our guidelines. A member of our team will be in touch shortly with a set of requests. As we can't proceed until these requirements are met, your swift response will help prevent delays to publication. Please also make sure to address the data and other policy-related requests noted at the end of this email.

*Copyediting*

*Published Peer Review History*

*Early Version*

*Submitting Your Revision*

Sincerely,

Gabriel Gasque, Ph.D.,

Senior Editor,

ggasque@plos.org,

PLOS Biology

ETHICS STATEMENT:

-- Please indicate within your manuscript if you experimental protocol was approved by your Institutional Review Board or a similar ethical committee.

-- Please include the ID number of the approved protocol.

DATA POLICY:

-- Please edit your README file in https://github.com/hrl-team/mfree_imitation to explain how your data was analyzed to generate the plots displayed in Figures Fig 3ab, 4, 5acd, 6ab, S1, S2, S3abc, S4, S5, S6., S7, S8, and S10.

-- In addition, please deposit spreadsheets with the individual numerical values that underlie the summary data displayed in the aforementioned graphs. The numerical data provided should include all replicates AND the way in which the plotted mean and errors were derived (it should not present only the mean/average values). For an example see here: http://www.plosbiology.org/article/info%3Adoi%2F10.1371%2Fjournal.pbio.1001908#s5

Reviewer remarks:

Reviewer #1: No additional comments

Reviewer #3: Thank you for addressing my previous comments. I think the changes have improved the manuscript and have no further comments.

---

## [Editor Report · Decision Letter 3]

10 Nov 2020

Dear Dr Palminteri,

On behalf of my colleagues and the Academic Editor, Matthew F. S. Rushworth, I am pleased to inform you that we will be delighted to publish your Research Article in PLOS Biology. 

PRODUCTION PROCESS

Before publication you will see the copyedited word document (within 5 business days) and a PDF proof shortly after that. The copyeditor will be in touch shortly before sending you the copyedited Word document. We will make some revisions at copyediting stage to conform to our general style, and for clarification. When you receive this version you should check and revise it very carefully, including figures, tables, references, and supporting information, because corrections at the next stage (proofs) will be strictly limited to (1) errors in author names or affiliations, (2) errors of scientific fact that would cause misunderstandings to readers, and (3) printer's (introduced) errors. Please return the copyedited file within 2 business days in order to ensure timely delivery of the PDF proof. 

If you are likely to be away when either this document or the proof is sent, please ensure we have contact information of a second person, as we will need you to respond quickly at each point. Given the disruptions resulting from the ongoing COVID-19 pandemic, there may be delays in the production process. We apologise in advance for any inconvenience caused and will do our best to minimize impact as far as possible.

EARLY VERSION

PRESS 

Kind regards,

Vita Usova

Publication Assistant, 

PLOS Biology

on behalf of

Gabriel Gasque,

Senior Editor

PLOS Biology